# On Magnetic Models in Wavefunction Ensembles

**DOI:** 10.3390/e25040564

**Published:** 2023-03-25

**Authors:** Leonardo De Carlo, William D. Wick

**Affiliations:** 1Scuola Normale Superiore, Piazza dei Cavalieri, 7, 56126 Pisa, Italy; 2Department of Economics and Finance, Luiss Guido Carli, Viale Romania, 32, 00197 Rome, Italy; 3Independent Researcher, Seattle, WA 98119, USA

**Keywords:** quantum magnetism, wavefunction ensembles, large deviations

## Abstract

In a wavefunction-only philosophy, thermodynamics must be recast in terms of an ensemble of wavefunctions. In this perspective we study how to construct Gibbs ensembles for magnetic quantum spin models. We show that with free boundary conditions and distinguishable “spins” there are no finite-temperature phase transitions because of high dimensionality of the phase space. Then we focus on the simplest case, namely the mean-field (Curie–Weiss) model, in order to discover whether phase transitions are even possible in this model class. This strategy at least diminishes the dimensionality of the problem. We found that, even assuming exchange symmetry in the wavefunctions, no finite-temperature phase transitions appear when the Hamiltonian is given by the usual energy expression of quantum mechanics (in this case the analytical argument is not totally satisfactory and we relied partly on a computer analysis). However, a variant model with additional “*wavefunction energy*” does have a phase transition to a magnetized state. (With respect to dynamics, which we do not consider here, wavefunction energy induces a non-linearity which nevertheless preserves norm and energy. This non-linearity becomes significant only at the macroscopic level.) The three results together suggest that magnetization in large wavefunction spin chains appears if and only if we consider indistinguishable particles and block macroscopic dispersion (i.e., macroscopic superpositions) by energy conservation. Our principle technique involves transforming the problem to one in probability theory, then applying results from large deviations, particularly the Gärtner–Ellis Theorem. Finally, we discuss Gibbs vs. Boltzmann/Einstein entropy in the choice of the quantum thermodynamic ensemble, as well as open problems.

## 1. Motivations, Results and Organization of the Work

In recent decades not only has entanglement been observed both at the microscopic level [1] and in “semi-macroscopic” objects [2,3,4] (even at room temperature [5]), but remarkably also superposition states became fundamental to describe not only atomic-level observations [6,7,8] but also large atom complexes [9,10,11,12], for which they observed “cat” states, i.e., states where the center of mass (COM) displayed spatial dispersion or interference patterns. These developments are leading to an experimental program to locate the classical-quantum boundary and the underlying mechanism [13].

In our view, the above-mentioned phenomena can be understood only by considering the wavefunction as a configuration of matter, encoding a physical state in a high-dimensional Hilbert space whose effect is felt in the ordinary three-dimensional space, as Schrödinger originally intended [14]. But this view also implies paradoxes, as pointed out by Schrödinger himself [15]: the unrestricted linearity of quantum mechanics produces situations where macroscopic objects exhibit COM dispersion. (Maybe this was the biggest handicap of his theory and view that particles are not individuals.) On the other side the unrestricted linearity of quantum mechanics is essential for many of the successful predictions at the atomic–nuclear level, but today there is a shared opinion [13] that quantum mechanics could have a size limit to its application. This idea and the mentioned observations about superposition states raise the possibility of deviations from Schrödinger’s equation. This research line was encouraged by S. Weinberg in a recent popular account [16] where he exposed the struggle of quantum mechanics with the macroscopic world known as the “measurement problem”. He explained that modifications have to be undetectable at atomic–nuclear level and at the same time “eliminate” macroscopic superpositions without granting the apparatus any special status. We mention that Weinberg in 1989 developed a mathematical framework [17] to test non-linear generalizations of Schrödinger’s equations at the atomic–nuclear level. Subsequent experiments appeared to rule out the models he examined [18]. (However, models with wavefunction energy escape this net [19].)

Here we consider wave-mechanical configurations for simple spin models with the goal of constructing Gibbs statistical ensembles of magnets containing all the states of the Hilbert space of quantum mechanics. We found it relevant to consider a modification (small for few degrees of freedom and very large for many degrees of freedom in a sense to clarify later) of quantum mechanics represented by an energy giving a “cost” to superpositions, that we call “wavefunction energy” (WFE), thus eliminating the macroscopic dispersion that afflicted the original wave mechanics of Schrödinger. The corresponding dynamical framework is the Hamiltonian one of Weinberg [17] but with a very important difference, observed in [19], that implies conservation of the norm during the evolution. This general set-up and the principles of this proposal will only be briefly described in Section 3.1 since dynamical problems are not in the purview of the present paper.

## 2. Materials and Mathematical Results

In standard quantum mechanics, one first diagonalizes the Hamiltonian to discover the eigenvalues; call them $\left\{E_{n}\right\}$; then, if the energy is ever measured, the result would be to “find” one of the $E_{n}$, and the system jumps into the corresponding eigenstate. (The realist assumption that a system actually exists in one of the eigenstates leads to contradictions whenever certain pairs of observables are involved, such as position and momentum, or energy and time. Hence the reliance on textual formulations involving “finding” rather than “being”. (John Bell made these points eloquently, especially in his last paper “Against ‘measurement”’ [20].)) Thus a thermal ensemble is constructed using these energy-measurement “outcomes” of form:(1)Fβ..=..Z−1∑nexp{−βEn/kT}<ψn|F|ψn>,
where β is the inverse temperature and $\psi_{n}$ is the eigenstate corresponding to eigenvalue $E_{n}$ in the Hilbert space. In the special case of the Heisenberg model, with Hamiltonian operator expressed in terms of Pauli matrices, where only the z-axis interacts $H=-\;2\;\sum_{i,j;n.n.}^{N}\;\sigma_{z;i}\;\sigma_{z;j}$, the model reverts to the Ising model: interpreting the classical spins $S_{i}$ as pointing up or down the z-axis and as labels of Hilbert space vectors, e.g., in Dirac notation, $|S_{1},\ldots ,S_{N}>$, they already define an eigenbasis of *H*.

In our first attempt to treat wave-mechanical spin models, we restrict to *N* spins $S_{1},\ldots ,S_{N}$ taking values $\pm \frac{1}{2}$ and assume interaction only along the z-axis. The state is not now a spin configuration but a wavefunction on such configurations, i.e., of form $\psi (S_{1},\ldots ,S_{N})$ where each value is a complex number. Thus ψ lies in a space of $2.2^{N}$ real dimensions (with the notation $2.2^{N}$ we mean $2^{N}$ dimensions because of the possible classical configurations and with $2.2^{N}$ we double the dimensions because of the real part and the complex one of a state ψ). The energy of a wavefunction ψ is what is called in standard quantum mechanics “the expected energy”. In particular, the energy E(ψ) contains contributions from all spin configurations (“superpositions”). The Gibbs canonical ensemble is not based on a product measure.

The probability generating function we wish to evaluate now takes the form:(2)ZN(β;λ1,…,λN)..=..∫||ψ||=1dψexp−βE(ψ)+ΛN(ψ),
where ${\left||\psi |\right|}^{2}=\sum_{S}\;{\left|\psi \left(S_{1},\ldots ,S_{N}\right)\right|}^{2}$ with *S* a classical configuration of spins. The integral is over the unit sphere in the above-stated number of dimensions. The energy is
(3)$$E_{N}\left(\psi \right)=-<\psi |\;E\left(S\right)\;|\psi >=-\sum_{S}\;{\left|\psi \left(S_{1},\ldots ,S_{N}\right)\right|}^{2}\;E\left(S\right),$$
with E(S) a classical spin magnetic energy of order *N* in the configuration *S*, as Curie–Weiss $\frac{1}{N}\;{\left(\;\sum_{i=1}^{N}\;S_{i}\;\right)}^{2}$ or Ising $\sum_{n.n.}S_{i}S_{j}$ (n.n. means nearest neighbor sites *i* and *j*), and
(4)$$\Lambda_{N}\left(\psi ;\lambda_{1},\ldots ,\Lambda_{N}\right)=<\psi |\sum_{i=1}^{N}\;\lambda_{i}\;S_{i}\;|\psi >=\sum_{S}\;{\left|\psi \left(S_{1},\ldots ,S_{N}\right)\right|}^{2}\sum_{i=1}^{N}\;\lambda_{i}\;S_{i}.$$
The sum $\sum_{S}$ in (Equation 3) and (Equation 4) is the sum over all spin configurations.

The motivation for the base integral over normalized wavefunctions differs from the Copenhagenist, in which ψ represents a probability distribution on spin configurations. Rather, Schrödingerist reasoning is that we do not want to compare states on the basis of normalizations, e.g., one with norm 0.1 and another with norm 100, but solely by their respective energies. Moreover, both linear quantum mechanics and suitable nonlinear generalization [19], preserve the norm, and the latter raises the possibility that the flow is ergodic or chaotic, with a possible interest in a classical justification for the Second Law of Thermodynamics. Thus we would not wish to drop the normalization.

The objective of this work is to understand whether it is possible to construct physically meaningful wave-mechanical magnetic models and to explore the implications. In Section 3 we consider distinguishable particles, namely we take the full $2.2^{N}$-sphere as state space without imposing any exchange symmetry on the wavefunctions. In this case the models will not magnetize (except at 0 temperature): $\lim_{N\to \infty }{\;[\;\{\;|m\left(\psi \right)|>\epsilon \;\}\;]}_{\beta }=0$ for any ϵ, where $m\left(\psi \right)\;=\;<\psi |\;\sum \;S_{i}{/N\;|\psi >\;=\;\sum \;|\psi \left(S\right)|}^{2}\left(\;\sum \;S_{i}\;\right)/N$ and ${\left[\cdot \right]}_{\beta }$ is the ensemble thermal average.

In Section 3.1 we introduce a generalized Hamiltonian framework, as developed in [17,19], assuming a non-linearity that penalizes large superpositions in large objects. This non-linearity makes energetically impossible configurations corresponding to macroscopic cat states. (For a modern mathematical definition of cat states see [19].) Even in this case, with distinguishable particles no phase transition appears (except at zero temperature).

In Section 4 we specialize to the mean field case with exchange symmetry. With this choice, the dimensionality of the Hilbert state space is greatly reduced and the model manifests magnetization at finite temperatures, when wavefunction energy is included. (In this case but without WFE, showing that there is no phase transition assumes some properties of a certain two-argument function for which we provide computer-generated evidence). To our knowledge, this is the simplest wavefunction model in which a phase transition appears.

In Section 5, we discuss the choice of ensembles with uniform measure, as opposed to other possibilities. In particular we introduce a base measure that includes a quantum Boltzmann–Einstein entropy and discuss how the usual ensembles should be recovered in a suitable limit.

The paper is organized as follows: in each section the primary results are described but the mathematical derivations are relegated to Appendix A. Our principle technique involves transforming the problem to one in probability theory, then applying results from large deviations, particularly the Gärtner–Ellis Theorem.

**Definition of phase transition.** In classical spin magnetic models one is interested in whether in the thermodynamic limit (N→∞) there is a spontaneous magnetization at sufficiently low temperature. This means the following. Define the magnetic field generated by the N spins $S_{i}$ as:(5)M..=..∑i=1NSi.
From the up–down symmetry of E(S) one deduced that ${\left[\;M\;\right]}_{\beta }=0$, so the average is not the relevant question. There are two ways to continue. One can introduce + boundary conditions, say by fixing $S_{1}=S_{N}=+\frac{1}{2}$, breaking the symmetry, and inquire as to whether, below a critical temperature,
(6)$$\lim_{N\to \infty }\;{\left[\;M\;\right]}_{\beta }>0.$$
Another approach is to maintain the “free” boundary conditions and inquire into the behavior of the field per spin, M/N; does it exhibit Central Limit Theorem (CLT) behavior; that is, ≈ O$\left(1/\sqrt{N}\right)$? This implies ${\left[\;|M|\;\right]}_{\beta }\approx \sqrt{N}$. If one interprets “O(N)” as macroscopic, one concludes that spontaneous magnetization did not appear. The CLT would hold if the correlations decayed exponentially:(7)$${\left[\;S_{i}\;S_{j}\;\right]}_{\beta }\approx \exp\left\{\;-\xi \;d\left(i,j\right)\;\right\};$$
(for some constant ξ>0, where d(i,j) denotes distance between the lattice points labeled *i* and *j*) but not if correlations decayed more slowly or even were bounded away from zero. In that case, assuming below a critical temperature the dispersion
(8)$${\left[\;{\left(\;M/N\;\right)}^{2}\;\right]}_{\beta }\approx \mathrm{constant}>0,$$
one would conclude that the limiting ensemble was a mixture of magnetized states and a phase transition had occurred. Observing the behavior (Equation 8) is the concept of phase transition we use throughout the paper, where we replace M/N and ${\left(M/N\right)}^{2}$, respectively, with m(ψ) and $m^{2}\left(\psi \right)$.

## 3. Models with Distinguishable Particles

The starting point, see Section A.3, is to replace the random point on the sphere in (Equation 2) with
(9)$$\psi \left(S\right)⟶\frac{\chi (S)}{\sqrt{{\sum \;|\chi \left(S\right)|}^{2}}},$$
where $\left\{\;\chi \left(S\right):S=\left(S_{1},\ldots ,S_{N}\right)\;\right\}$ are $2^{N}$ complex, or 2.$2^{N}$ real, numbers distributed as i.i.d. standard (mean zero, norm one) Gaussians. This procedure defines a measure on the sphere and the rotational symmetry of the standard Gaussian measure (covariance matrix: the identity) then assures that we are considering the same distribution of wavefunctions. By this transformation and various lemmas, we will reduce the problems of next sections to finding probabilities of rare events (called “large deviations” theory). Let
(10)m(ψ)=<ψ|∑Si/N|ψ>=∑|ψ(S)|2(∑Si/N);[F(ψ)]β..=..∫||ψ||=1dψexp{−βEN(ψ)}F(ψ)/ZN.

**Theorem 1.** 
*Given a magnetic spin energy E(S) of order N, e.g., Curie–Weiss or Ising, in any dimension for any finite temperature (except T = 0) there is no magnetization in the thermodynamic limit:*

(11)
$$\lim_{N\to \infty }{\;[\;\{\;|m\left(\psi \right)|>\epsilon \;\}\;]}_{\beta }=0,$$

*for any ϵ>0.*


For the proof, see Math Section A.3. This case does exhibit zero-temperature magnetization. Consider Curie–Weiss or Ising and β→∞, the probability distribution becomes concentrated on states of minimal energy. These have the form:(12)ψ(θ,α)=cos(θ)ψ+..+..sin(θ)e−1αψ−,
where $\psi_{\pm }$ denotes a wavefunction concentrated on all up, respectively all down, spins. Hence at T=0 the magnetization takes the form:∫0π/2dθf(θ)<ψ(θ,α)|MN|ψ(θ,α)>2..=..14∫0π/2dθf(θ)cos2(2θ)>0.

### 3.1. Models with Wavefunction Energy

We consider a modification of the $E_{N}\left(\psi \right)$ adding nonquadratic terms into the wavefunction. We will refer to these terms as representing “wavefunction energy” (WFE). Our principles to modify the usual quantum Hamiltonian are:(a)The modification has to be negligible at the microscopic level but becomes large enough at the macroscopic level to block dispersion (macroscopic dispersion of the spins is better explained later);(b)The norm of ψ and the energy of a closed system have to be conserved (the same for momentum);(c)No extra terms are added to the evolution of the center of mass $X\left(\psi \right)=<\psi |\frac{1}{N}\sum_{N}^{j=1}x_{j}|\psi >$ of a closed system.

These characteristic are satisfied when the evolution of the quantum state is
(13)$$i\hbar \frac{\partial \psi }{\partial t}=\frac{\partial }{\partial \psi^{*}}E\left(\psi \right).$$
with
(14)E(ψ)..=..EQM(ψ)..+..EWFE(ψ),
(15)$$E_{WFE}\left(\psi \right)=wN^{2}D\left(\psi \right),\;\;D\left(\psi \right):=<\psi \left|\;{\left(\;\overset{N}{\sum_{i=1}}\;O_{i}-<\psi \left|\;\overset{N}{\sum_{i=1}}\;O_{i}\;\right|\psi >\;\right)}^{2}\;\right|\psi >$$
where *w* is a very small constant and the $O_{i}$’s are self-adjoint operators diagonal in the same base as $H_{QM}$, see the proofs in [19]. When in (Equation 14) we take w=0 the energy E(ψ) becomes $\langle \psi |H_{QM}|\psi \rangle$ where $H_{QM}$ is the usual quantum Hamiltonian given by the sum of the kinetic and potential terms; taking the derivative with respect to $\psi^{*}$ in (Equation 13) one finds the Schrödinger’s equation.

In [19,21] one author explored nonlinear Schrödingerist quantum mechanics as a potential solution to the Measurement Problem: the addition of nonquadratic terms to the Hamiltonian, i.e., the WFE, were proposed to block spatial dispersion in macroscopic objects (and otherwise be too small to matter). The WFE in the present work will be a functional of the wavefunction addressing dispersion of the total spin, instead of center of mass or total momentum (as in [22]). This means $O_{i}=S_{i}$. This choice just reflects the lack of spatial coordinates in our models of magnetism. The straightforward generalization of (Equation 16) to models with spatial coordinates is $O_{i}=L_{i}+S_{i}$ in (Equation 15) and [*w*] = kg${}^{-1}$m${}^{-2}$. The original proposal [23] is on the momentum $P_{i}$. The general idea behind (Equation 15) derived in part from an interview with Hans Dehmelt, see chapter 15 in [24]. He explained to one of the authors that classical physics applies when for some reason wavefunctions cannot spread out (forming cats). For instance, classical-like orbits in electromagnetic traps are due to the trapping fields. The mechanism (Equation 15) proposes a universal self-trapping forbidding macroscopic dispersion, so to speak. (The ideal experimental test to falsify WFE is described in [22]).

For our statistical ensembles the first term $E_{QM}\left(\psi \right)$ now incorporates the usual spin–spin and spin–external field interactions, while the second $E_{WFE}\left(\psi \right)$ takes the form:(16)EWFE(ψ)..=..wN2D(ψ);D(ψ)..=..1N2<ψ|∑Si−<ψ|∑Si|ψ>2|ψ>..=..1N2<ψ|∑Si2|ψ>−<ψ|∑Si|ψ>2.
Here *w* is a positive constant small enough to make the mechanism undetectable at atomic level and we have written the sum of spins rather than *M* to emphasize the interpretation as the “dispersion of the center-of-spin”. We observe that D(ψ) is of order 1 on *cat states*, i.e., close to (the concept can be made precise introducing a suitable spherical distance)
(17)$$\psi =\frac{1}{\sqrt{2}}\psi_{+}+\frac{e^{\sqrt{-1}\alpha }}{\sqrt{2}}\psi_{-},$$
and therefore $E_{WFE}\left(\psi \right)$ of order $wN^{2}$, of order 1/N on *product states*, i.e., close to
(18)P.S...=..ψ:…ψ(S1,…,SN)=∏i=1Nψi(Si),|ψi(1/2)|2+|ψi(−1/2)|2..=..1,…∀i.
and therefore $E_{WFE}\left(\psi \right)$ of order wN, of order $1/N^{2}$ close to *eigenstates* (classical configurations with no superpositions) and therefore $E_{WFE}\left(\psi \right)$ of order *w*. We observe that for the macroscopic magnetization M(ψ)=Nm(ψ) the situation will be opposite, that is it will be very small close to product and cat states while of order *N* close to eigenstates.

Let us first examine T=0. Again, the case reduces to support on wavefunctions of form:(19)ψ(θ,α)=cos(θ)ψ+..+..sin(θ)e−1αψ−,
plugging into the definition of *D* yields:(20)$$E_{WFE}\left(\psi \right)=\frac{N^{2}}{4}\;\sin^{2}\left(2\;\theta \right).$$
Since by adding a trivial constant (which does not alter the measure) the usual energy, respectively, for Ising and Curie–Weiss, can be rewritten:(21)$$E_{QM}\left(S_{1},\ldots ,S_{N}\right)=\sum_{i,j,n.n.}\;{\left(\;S_{i}-S_{j}\;\right)}^{2}\;\mathrm{and}\;N\left(1-m^{2}\left(S\right)\right),$$
both forms of energy have the same sign (positive). So in the presence of the WFE terms the support is reduced to the cases with θ=0 or θ=π/2; hence, the ensemble becomes a mixture of two magnetized states, as in (Equation 13). “Classical” behavior is restored.

Thus, the interesting question arises as to whether this conclusion will persist at nonzero temperatures. We observe that the “correlations” suppressed by including $E_{WFE}\left(\psi \right)$ in the Gibbs factor concern single wavefunctions and whether they imply a cat, while the correlations important for magnetization are thermal. For a discussion about why these models can not be regarded as “continuous spin” models, see [25].

In Math Section A.3 we show that adding the WFE to distinguishable particles does not change the conclusion about magnetization (namely, none at any finite temperature).

**Proposition 1.** *For distinguishable particles with Hamiltonian* (Equation 14)*, given a magnetic spin energy E(S) of order N, e.g., Curie–Weiss or Ising, in any dimension for any finite temperature (except T = 0) there is no magnetization in the thermodynamic limit:*
(22)$$\lim_{N\to \infty }{\;[\;\{\;|m\left(\psi \right)|>\epsilon \;\}\;]}_{\beta }=0,$$
*for any ϵ>0.*

Namely, even under non-linear modifications, it is not possible to produce phase transitions, which we attribute to entropy swamping energy because of high dimensionality of the models.

## 4. Models Assuming Exchange Symmetry: The Schrödingerist Curie–Weiss Model

As starting point to remove the notion of distinguishable particles we consider the Curie–Weiss energy:(23)ECW(S)..=..−1N∑i=1NSi2,
where we are going to replace spin configurations by wavefunctions with exchange symmetry; for such a system one should consider “integer spin”, say {−1,0,+1}; see [26,27] for examples of integer spin models. Here we adopt two to learn how to construct some mathematical tools for these wave-mechanical models, reasoning that the lowest possible dimensionality and mean field interaction define the first case to study whether phase transitions appear or not. Models with higher dimensions, meaning higher entropy, or local interactions presumably are even less likely to exhibit such transitions. (Both the concept and the mathematical definition of nearest neighbors for indistinguishable particles appears somewhat problematic in wavefunction models.) In [28], qu-bits are considered with the exchange symmetry. IBM [29] (see references there) says that titanium atoms can represent qu-bits and they project to make a complex of many of them to observe how the collective behavior changes; this might be a situation described by the SCW model.

We define a wave-mechanical model with Hamiltonian (Equation 23) and exchange spin symmetry. Note that with two levels the CW energy depends only on the number, call it “*n*”, of “down” spins. Thus we are led to introducing wavefunctions that depend only on “*n*”. On N-dimensional Euclidean space, define, given some $a_{i}>0$:(24)∫{∑xi2ai=1}∏i=1NdxiF(x1,…,xN)..=..limσ→0Aellipsoid−1∫∏dxiexp{−(∑xi2ai−1)2/σ2}F(x1,…,xN)/W(σ),
where “$A_{\mathrm{ellipsoid}}$” stands for the surface area of the ellipse $\left\{\;\sum \;x_{i}^{2}\;a_{i}=1\;\right\}$ and
(25)W(σ)..=..∫∏dxiexp{−(∑xi2ai−1)2/σ2}.

Thus we utilize an approximate Dirac delta-function becoming concentrated in the limit on the ellipsoid. Now let us make a change of variables by defining: $y_{i}=x_{i}/\sqrt{a_{i}}$ in (Equation 24). A Jacobian derivative of $\prod \sqrt{a_{i}}$ appears in numerator and denominator and so cancels, yielding:

**Lemma 1.** 
*Let $A_{\mathrm{sphere}}$ be the “surface area” of the N-sphere. Then:*

∫{∑xi2ai=1}∏i=1NdxiF(x1,…,xN)..=..AsphereAellipsoid∫{∑yi2=1}∏i=1NdyiF(a1yi,…,aNyN).



We want to define an ensemble and corresponding integrals for symmetric wavefunctions. Each such wavefunction is uniquely given by its common value, call it $\hat{\psi }\left(n\right)$, on the set of configurations with *n* “down” spins. We have then:$$\begin{array}{ccc} {\left||\psi |\right|}^{2} & = & \sum_{n=0}^{N}\;C\left(N,n\right)\;{\left|\hat{\psi }\left(n\right)\right|}^{2}=1,\\ E(\psi ) & = & \sum_{n}\sum_{\left\{\#S=n\right\}}{\left|\psi \left(S\right)\right|}^{2}\;E_{S}=\sum_{n}C\left(N,n\right){\left|\hat{\psi }\left(n\right)\right|}^{2}E_{n},\;\mathrm{where}\;E_{n}:=\sum_{\left\{\#S=n\right\}}E_{S}/C\left(N,n\right). \end{array}$$
Here “*S*” stands for a spin configuration, “#S” for how many “down” spins it contains and C(N,n) is the number of combinations with *n* spins down. Next we let
(26)$$\phi \left(n\right):=\sqrt{C(N,n)}\;\hat{\psi }\left(n\right),$$
and apply the theorem from above section, yielding:(27)$$\int_{\left\{\;\sum \;|\hat{\psi }\left(n\right){|}^{2}=1\;\right\}}\;\exp\left\{-\;\sum \;C\left(N,n\right)\;|\hat{\psi }{\left(n\right)|}^{2}\;E_{n}\;\right\}=\frac{A_{\mathrm{sphere}}}{A_{\mathrm{ellipsoid}}}\;\int_{\left\{\;\sum \;|\phi \left(n\right){|}^{2}=1\;\right\}}\;\exp\left\{-{\;\sum \;|\phi \left(n\right)|}^{2}\;E_{n}\;\right\}.$$
Note that the prefactor in (Equation 27) is not in the integrand, not a function of ϕ, just a constant depending on *N*. Thus it plays no role in computing, e.g., the magnetization. With these preparatory remarks we can define our symmetric-wavefunction SCW (Schrödingerist Curie–Weiss) model. The magnetic energy associated with ϕ becomes:(28)ECW(ϕ)..=..−1N∑n=0N|ϕn|2N−2n2.
Our SCW model is then defined by:$$\begin{array}{c} {\left[\;F\left(\phi \right)\;\right]}_{\beta }=\int_{\left\{||\phi |{|}^{2}=1\right\}}d\phi \;\exp\left\{-\beta \;E_{CW}\left(\phi \right)\right\}\;\frac{F(\phi )}{Z},\;\;\;Z=\int_{\left\{||\phi |{|}^{2}=1\right\}}\;d\phi \;\exp\left\{-\beta \;E_{CW}\left(\phi \right)\right\}. \end{array}$$
The choice of the uniform base ensemble might be questioned; we leave this issue to Section 5.

### 4.1. The SCW Model with Wavefunction Energy

We observe that
(29)ECW(ϕ)..=..−Nm2(ϕ)+D(ϕ),
where
m(ϕ)=1N∑|ϕn|2N−2n;D(ϕ)=1N2∑|ϕ|2N−2n2..−..∑|ϕn|2N−2n2.
In passing, we point out the curious fact, revealed by (Equation 29), that D(ϕ) plays a role in a wavefunction model with the standard energy. We define:(30)f..=..Nβ1−m2(ϕ)−D(ϕ)+NwD(ϕ).
Here we have added a term (Nβ) to make f≥0 and incorporated wavefunction energy. The intuition for the latter choice comes from the observation that, lacking that term, *f* can be small if *either*
$m^{2}$ is large or *D* is large; incorporating the dispersion term with large enough wN, the last possibility should be suppressed.

As always, the model is defined by:$$\begin{array}{c} {\left[\;m^{2}\;\right]}_{\beta }=\int_{{\left||\phi |\right|}^{2}=1}\;d\phi \;\exp\left\{-f\left(\phi \right)\right\}\;m^{2}\left(\phi \right)/Z;\;\;\;\;Z=\int_{{\left||\phi |\right|}^{2}=1}\;d\phi \;\exp\left\{-f\left(\phi \right)\right\}. \end{array}$$ Intuition suggests we should investigate cases were *w* is at least 1/N; hence, we define
(31)ω..=..Nw;
and we assume below that ω is a constant. This does not indicate a belief that *w* actually scales with *N*; if a wavefunction energy exists, then *w* is a constant of nature and does not scale. The role of the assumption is to avoid the suppression of all superpositions, as it would follow with fixed “*w*” in the mathematical limit of large *N* because of the factor $N^{2}$. This limit does not exist in reality; therefore our assumption is just a mathematical stratagem to prove theorems.

In Math Section A.4 we prove:

**Theorem 2.** 
*Let positive numbers ω and ϵ satisfy:*

(32)
$$\begin{array}{ccc} & & 1<\omega <4/3; \end{array}$$


(33)
$$\begin{array}{ccc} & & \epsilon <\frac{1}{4}\;{\left(\;1+\sqrt{1-4\;r}\;\right)}^{2}; \end{array}$$


(34)
r..=..ω−1ω.

*Then there is a positive number $\beta_{c}=\frac{p^{*}\left(\epsilon \right)}{(\omega -1)\epsilon }$ such that, for $\beta >\beta_{c}$,*

(35)
$$\lim_{N\to \infty }\;{\left[m^{2}\right]}_{\beta }>\epsilon .$$



The factor $p^{*}\left(\epsilon \right)$ is a large deviation rate functional related to the application of the Gärtner–Ellis theorem; we do not write it here because it involves a technical discussion on large deviations theory, so we prefer to postpone it to Math Section A.4. The bound 4/3 is required for the application of the Gärtner–Ellis theorem but we expect that it has no meaning. With the specified range for ω, (Equation 33) always holds if ε<1/4.

The fact that at high temperature the magnetization is zero is in Section A.2, where we used a system of coordinates called polyspherical coordinates for a computation, see [30].

### 4.2. The SCW Model without Wavefunction Energy

Without WFE, we state this theorem:

**Theorem 3.** 
*For the SCW model at any finite temperature (except T = 0) and for any ϵ>0:*

(36)
$$\lim_{N\to \infty }{\;[\;\{\;|m\left(\phi \right)|>\epsilon \;\}\;]}_{\beta }=0.$$



Thus, the SCW model with the standard energy expression has no finite-temperature phase transition. Using large deviation techniques from probability theory, we reduced the proof of Theorem Three to a study of one real function of two variables, for which we supply exact formulas (see Math Section A.5). A property for the rate functionals, needed for the proof, is sustained by mathematical arguments and can be displayed, with some computational limits, via computer using statistical sampling from the domain of this function.

## 5. Ensemble Choice and Discussion

In previous sections, we constructed Gibbs ensembles where each state is weighted with a uniform measure on the sphere. A priori, we could consider other measures conserved by the Hamiltonian flows, e.g. modifications of the uniform one as dϕF(ϕ) where F(ϕ) satisfies {F(ϕ),E(ϕ)}=0 and the parenthesis {·} indicates the Poisson bracket. In the quadratic case this will be equivalent to [F,H]=0, where now [·] denotes the commutator. (This reflects the fact that there are many conserved quantities other than the energy and the norm in linear QM (including the moduli of the wavefunction component in the eigenstate directions), showing that linear QM is far from being ergodic.) As discussed in [19], Schrödinger’s equation and the non-linear modifications proposed there are simply Hamilton’s systems disguised; hence, Liouville’s theorem applies, implying that the base measure should be $\prod_{S}d\psi \left(S\right)$. The non-linear modification of WFE of course will eliminate the conserved quantities along eigenstate directions. Proving the ergodicity for some such modification would qualify the uniform measure as the unique choice for a base measure. But at the present time we can only demonstrate the existence of expanding and contracting directions in certain special cases, see [21].

An alternative ensemble, with the same base measure, could be constructed by adopting a Boltzmann factor that weights a state ψ taking into account the notion of possible microscopic states compatible with a macroscopic one. Concerning quantum theory, this notion can be tracked back to Einstein [31], in a paper about how to define quantum entropy. We think that, if the introduction of this modification has a meaning, it should be related to having considered indistinguishable particles jointly with a lack of spatial coordinates, which would add a degeneration on energy levels. For example, confining each spin in a spatial potential, we would have many levels for each spin and correspondingly many ways to arrange *n* spins down. In the SCW case, the proposal would be to replace in $\prod_{n}d\phi \left(n\right)$ with $\prod_{n}{\left[C\left(N,n\right)\right]}^{|\phi_{n}{|}^{2}}d\phi_{n}$, where C(N,n) is the number of states with *n* spins down. In this way, a wavefunction acquires a large combinatorical weight if amplitudes are concentrated on highly degenerated components.

Calling H the Hilbert space of wavefunctions of norm one decomposed as $H=\underset{n}{⊕}H_{n}$, where $H_{n}$ is the subspace of the wavefunctions with quantum number *n* and dimension $\dim H_{n}:=C\left(N,n\right)$, the proposed measure can be written also as
(37)$$\prod_{n}{\left[\dim H_{n}\right]}^{|\phi_{n}{|}^{2}}d\phi_{n}=e^{\left[\log\dim H](\phi \right)}\prod_{n}d\phi_{n},$$
with $\left[\log\dim H\right]\left(\phi \right)=\sum_{n}{\left|\phi_{n}\right|}^{2}\log\left(\dim H_{n}\right)$ and it is verified $\{\left[\log\dim H\right]\left(\phi \right),H_{SCWM}\left(\phi \right)\}=0$. Whether such a modified measure is still conserved by the Hamiltonian flow, with or without WFE, will depend on the precise details of the dynamics.

An a priori weight associated with a single wavefunction is questionable. However, at the moment we cannot exclude a different measure from the uniform one on the state space. Let us reflect on how this ensemble would introduce an analogous notion to the classical microcanonical entropy of the usual spin models. Introducing the energy and entropy densities, respectively,
$$e\left(\phi \right)=\langle \phi |e|\phi \rangle ,\;\mathrm{where}\;e\left(n\right)=-m_{N}^{2}\left(n\right)=-{\left(1-\frac{2n}{N}\right)}^{2},$$
$$s\left(\phi \right)=\frac{1}{\beta }\langle \phi |s|\phi \rangle ,\;\mathrm{where}\;s\left(n\right)=\frac{1}{N}\log C\left(N,n\right).$$
By Sterling, for large *N*, s(n) is approximated by
$$s\left(m_{N}\left(n\right)\right)=-\frac{1-m_{N}\left(n\right)}{2}\log\frac{1-m_{N}\left(n\right)}{2}-\frac{1+m_{N}\left(n\right)}{2}\log\frac{1+m_{N}\left(n\right)}{2}.$$
We also introduce
$$f^{\beta }\left(\phi \right):=e\left(\phi \right)-s\left(\phi \right).$$
The partition function becomes
(38)$$\begin{array}{c} \frac{1}{S_{N}}\int_{∥\phi ∥=1}e^{\log\dim H_{n}\left(\phi \right)}d\phi \;\;\exp\left(-\beta E\left(\phi \right)\right)\approx \frac{1}{S_{N}}\int_{∥\phi ∥=1}d\phi \;\;e^{-\beta Nf^{\beta }\left(\phi \right)}. \end{array}$$
Since we are interested in making *N* large, we used the approximation $\;\frac{1}{\beta N}\log\dim H_{n}\left(\phi \right)\approx s\left(\phi \right)$ and, neglecting the errors, we arrive at
(39)$$\begin{array}{c} Z_{N}=\frac{1}{S_{N}}\int_{∥\phi ∥=1}d\phi \;\;e^{-\beta N[-{\left(m\left(\phi \right)\right)}^{2}-s\left(\phi \right)-D\left(\phi \right)]}. \end{array}$$
The extra factor $e^{\beta Ns(\phi )}$ can be thought of as trying to restore a classical picture of a competition between internal energy and entropy due to high degeneracy of disordered macrostates. Anyway, since the role of this factor is giving large weights to some states of zero magnetization the WFE will be still necessary to define magnetic models.

The way to recover the usual ensembles may be to introduce $wN^{2}\;D$ with *w* constant (i.e., without keeping ω=wN constant) in the thermodynamic limit. In this case, one expects that the measure will concentrate on the set {ϕ:D(ϕ)=0}. Conceivably, we might arrive at the classical ensembles even in models without exchange symmetry, as considered in Section 3. However, apart from conflicting with a fundamental fact of quantum mechanics, i.e., that particles are indistinguishable and so wavefunctions must have symmetries, the ensemble will not magnetize in the thermodynamic limit even with suppression of superpositions, as already discussed after (Equation 22). We mention this is the route so far followed by various authors [32,33,34,35,36], where they reduced the ensembles on the sphere to the usual ones by introducing delta measures on eigenstates by different reasoning. The concentration we expect to be true for indistinguishable particles.

An interesting computation could be repeating the analysis of Theorem 3 with the modified measure (Equation 37), keeping first ω constant and in a second step considering the critical temperature with ω=wM in the limit of large *M*.

We guess that the uniform measure is still the right choice. The introduction of the Boltzmann factor appears a bit artificial, but it seems relevant when including $wN^{2}\;D$ with *w* constant, as otherwise the internal energy would have no competition, assuming the measure concentrates on eigenstates. However, we suggest there could be other meaningful pictures.

## 6. Conclusions

Once indistinguishable particles are considered, it seems we are forced to use a mean field type of interaction (plus the non-quadritic term of WFE to observe magnetization) because of the symmetry conditions on the wavefunctions at the moment we do not see a way to implement interactions such as nearest neighbors. (Current literature implementing nearest neighbor interactions on quantum states does not help, since they seem to apply the Hamiltonians without considering symmetry conditions.) To enrich our models we could consider to study the mean field for the Heisenberg model or considering the present model with a transverse magnetic field, which might be of interest as a model to study what are called quantum phase transitions [37]. Developing these ensembles could be of help to deal with problems for continuous models related to diffusion.

We might interpret the necessity of considering wavefunctions with symmetry conditions if we hope to have a thermodynamics that includes phase transitions as another justification of the usual quantum rules for combining indistinguishable systems.

Assuming the validity of Theorem Three (which claims that SCW with the usual energy does not spontaneously magnetize), our results suggest that blocking magnetic cats is related to phase transitions, presumably because states which are broad superpositions tend to have low magnetizations.

Perhaps the two main tasks for future investigations (which appear already very hard) are: improving the ability to compute spherical integrals, in the hope of deriving the functional dependence of magnetization on temperature; and generalizing the model to the case {−1,0,1} for symmetric wavefunctions and to the case {−1,1} for antisymmetric wavefunctions. We observe that these cases will automatically add degeneration to energy levels.

Also it might be worth considering *N*-levels for each spin, namely a pictorial way for replacing the discrete lattice with a confining potential for each spin and studying the limit for $wN^{2}$; this might help an understanding of Boltzmann entropy from a quantum perspective.

## Data Availability

The work did not need data.

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
