# Peer review of "On Magnetic Models in Wavefunction Ensembles"

_entropy, 2023, doi:10.3390/e25040564_

Round 1

Reviewer 1 Report

This paper, as I understand it, represents an attempt to mathematically reduce some quantum effects from the configuration space where these naturally “live” to the 3D space where the physical measurement takes place. On the basis of an appropriate spin model the authors manage to achieve their goal although it is not clear whether the results are applicable to more general problems. It would be therefore more beneficial for the readers if in the Conclusion the authors comment on the range of  applicability of their results instead of talking about Theorems, etc.

Author Response

I add a common file, since it can be of help for everyone.

Reviewer 2 Report

The paper studies the possibility of a phase transition (magnetization) at the thermodynamic limit in spin models with and without nonlinear modification of quantum mechanics. It is shown that the phase transition takes place in the case of nonlinear modifications (preventing macroscopic coherent superpositions) in the case of zero temperature and in the case of non-zero temperatures but indistinguishable particles. Some theorems are proved purely rigorously, one theorem uses a numerically tested conjecture. The results are interesting from both conceptual (limitations of quantum-mechanical description) and mathematical point of views. The manuscript can be recommended for publication.

The authors can take into account the following comments:

1. I think it is worth to explain in more detail what is meant here by 'phase transition' (for convenience of a broader audience, which might not be familiar well with this topic). Is it true that we speak about a phase transition if and only if we observe a magnetization at the thermodynamic limit?

2. It seems that the first section "1. How to Use this Template" (lines 21-27) should be removed.

3. Eq. (3): (a) In the middle expression, we see a summation over i, but the rest of the expression does not depend on i; (b) Why does the summation here start from i=2 rather than i=1? (c) What is the summation on the third expression? For convenience of a broader audience, it is worth to write these expressions in more detail to clarify these issues.

4. Eq. (9): It is worth to explain what is E_QM(psi) and how the standard Schroedinger equation is restored if E_WFE=0.

5. LIne 177: Pure states are defined as the states corresponding to classical configurations with no superpositions. However, in the standard axiomatics of quantum mechanics, pure states are arbitrary unit vectors in a Hilbert state (wave functions), in contrast to mixed states, which are density operators. Is it good to use this well-established notion in another meaning?

6. Line 193: Wrong citation reference ([?])

7. Lines 99 and 127: 2.2^N. Probably, 2\times 2^N is meant rather than 2.2^N (i.e. (2+2/10)^N).

Author Response

We upload a file with both replies, since we think it can be useful.

Round 2

Reviewer 2 Report

The authors successfully solved all the issues.